# The gastrointestinal pathogen *Campylobacter jejuni* metabolizes sugars with potential help from commensal *Bacteroides vulgatus*

Jolene M. Garber[1,2], Harald Nothaft[3], Ben Pluvinage[4], Martin Stahl[5,8], Xiaoming Bian[1,2], Sara Porfirio[2], Amber Enriquez[1,9], James Butcher [5], Hua Huang[6,10], John Glushka[2], Eric Line[7], John A. Gerlt[6], Parastoo Azadi[2], Alain Stintzi[5], Alisdair B. Boraston[4] & Christine M. Szymanski [1,2]*

Although the gastrointestinal pathogen *Campylobacter jejuni* was considered asaccharolytic, >50% of sequenced isolates possess an operon for L-fucose utilization. In *C. jejuni* NCTC11168, this pathway confers L-fucose chemotaxis and competitive colonization advantages in the piglet diarrhea model, but the catabolic steps remain unknown. Here we solved the putative dehydrogenase structure, resembling FabG of *Burkholderia multivorans*. The *C. jejuni* enzyme, FucX, reduces L-fucose and D-arabinose in vitro and both sugars are catabolized by *fuc*-operon encoded enzymes. This enzyme alone confers chemotaxis to both sugars in a non-carbohydrate-utilizing *C. jejuni* strain. Although *C. jejuni* lacks fucosidases, the organism exhibits enhanced growth in vitro when co-cultured with *Bacteroides vulgatus*, suggesting scavenging may occur. Yet, when excess amino acids are available, *C. jejuni* prefers them to carbohydrates, indicating a metabolic hierarchy exists. Overall this study increases understanding of nutrient metabolism by this pathogen, and identifies interactions with other gut microbes.

[1] Department of Microbiology, University of Georgia, 527 Biological Sciences Bldg, Athens, GA 30602, USA. [2] Complex Carbohydrate Research Center, University of Georgia, 315 Riverbend Road, Athens, GA 30602, USA. [3] Department of Biological Sciences, University of Alberta, 11455 Saskatchewan Drive, Edmonton, AB T6G 2E9, Canada. [4] Department of Biochemistry & Microbiology, University of Victoria, 3800 Finnerty Road, Victoria, BC V8P 5C2, Canada. [5] Ottawa Institute of Systems Biology and Department of Biochemistry, Microbiology and Immunology, University of Ottawa, 451 Smyth Road, Ottawa, ON K1H 8M5, Canada. [6] Institute for Genomic Biology, University of Illinois, 600 S Mathews Ave, Urbana, IL 61801, USA. [7] United States Department of Agriculture National Poultry Research Center, 950 College Station Rd, Athens, GA 30605, USA. [8] Present address: Division of Gastroenterology, BC Children's Hospital, Child and Family Research Institute and the University of British Columbia, 938 West 28th Avenue, Vancouver, BC V5Z 4H4, Canada. [9] Present address: Pinnacle Transplant Technologies, 1125 W Pinnacle Ave, Phoenix, AZ 85027, USA. [10] Present address: School of Life Sciences, South China Normal University, Qingsong W Rd, Guangzhou 510631, P. R. China. *email: cszymans@uga.edu

*C*ampylobacter jejuni is a common commensal in chickens and often transmitted to humans through consumption of undercooked or contaminated food products where it causes gastrointestinal infections that are generally self-limiting or treatable with antibiotics. However, post-infectious complications including Guillain-Barré syndrome, irritable bowel syndrome, reactive arthritis[1], and growth stunting[1,2] can occur. Additionally, increasing rates of *C. jejuni* antibiotic resistance, particularly to fluoroquinolones, is a growing concern[3].

*C. jejuni* was once considered asaccharolytic since it lacks key enzymes from the Entner-Doudoroff and pentose phosphate pathways for carbohydrate metabolism. Most of its nutrition is derived from the amino acids serine, aspartate, asparagine, glutamate, and proline, which are readily abundant in the chicken gastrointestinal tract[4]. Additionally, some, but not all strains can utilize glutamine[4,5]. Tricarboxylic acid cycle intermediates and short chain fatty acids regularly found in the gut as by-products of microbial metabolism are also used by *C. jejuni*[4,5] and their spatial distribution may be important for niche establishment in both commensal and pathogenic systems[6].

Carbohydrate utilization pathways have recently been described in certain *Campylobacter* species. For example, few isolates of *C. coli* and *C. jejuni* subsp. *doylei* possess enzymes for the full Entner-Doudoroff pathway and thus metabolize glucose[7,8]. Also, we identified a functional *fuc* locus for L-fucose utilization (*cj0480c-cj0489* in *C. jejuni* NCTC11168) that exists in ~65% of sequenced strains[9,10]. Many structures in the gastrointestinal tract contain L-fucose, including mucins[11], blood group antigens[12], the capsules, and glycoproteins of microbial species such as *Bacteroides*[13], dietary plant polysaccharides[14], and human milk oligosaccharides[15] in breastfed infants. L-fucose plays an important role in the health of the host and the maintenance of the associated microbiota, including microbiota-mediated resistance to pathogens[16], protection from Crohn's disease[17], and as a functional receptor for cholera toxin[18]. L-fucose is also an important nutrient for the microbiota, whether the bacteria can obtain it directly from the host[17,19,20] or rely on scavenging from their neighbors[21]. Furthermore, fucosylated structures can be important adherence targets[17], which has led to several studies reporting that human milk oligosaccharides may protect against infection by acting as binding decoys[15]. *Bacteroides* spp. are known to secrete fucosidases and can influence host fucosylation to maximize fucose utilization for nutrition[19]. However, *C. jejuni* does not encode any obvious fucosidase homologs suggesting that *C. jejuni* may forage on L-fucose released by other members of the microbiota[22,23] as has been observed for *Salmonella enterica* serovar Typhimurium, *Clostridium difficile*[24], and enterohemorrhagic *Escherichia coli*[21].

L-fucose enhances *C. jejuni* growth in minimal media and provides the strain with a colonization advantage in a piglet model of diarrheal disease[9]. Furthermore, we showed that strains possessing the *fuc* locus (*fuc*+) strains chemotax towards fucose, and the putative dehydrogenase Cj0485 is necessary for this response[25]. The fucose permease encoded by *cj0486* (*fucP*) complements an *Escherichia coli fucP* mutant[9]. However, L-fucose metabolism in *C. jejuni* does not rely upon ATP or GTP[9]. Thus, in *C. jejuni*, L-fucose must be catabolized by a pathway different from the common phosphorylation-dependent pathways found in *E. coli*, *Bacteroides thetaiotaomicron*[19], *Lactobacillus rhamnosus*[26], *S.* Typhimurium, and *Klebsiella pneumoniae*[27] and it is unlikely that L-fucose is used in *C. jejuni* capsular polysaccharides via the GDP-activation-dependent mechanism described for *Bacteroides*[28].

This work examines the crosstalk that may occur between commensal microbes and *C. jejuni* to enable L-fucose scavenging and provides evidence that *C. jejuni* can metabolize L-fucose released by commensal *Bacteroides vulgatus*. We also characterize the putative dehydrogenase Cj0485 through crystallography, enzymology, and multiple biological assays to confirm its function as a fucose dehydrogenase, which we have named FucX. This enzyme is also capable of reducing D-arabinose and is the sole component encoded by the *fuc* operon necessary for chemotaxis to both sugars. Nuclear magnetic resonance (NMR) and mutagenesis studies provide further insight into the mechanism of L-fucose breakdown and allowed us to propose a pathway for L-fucose and D-arabinose catabolism. We also investigate the impact of L-fucose on carbon source preferences by *C. jejuni*. Overall this study provides insight into the use of carbohydrates by *C. jejuni* and how the pathogen obtains sugars from the commensal microbiota providing options for future treatment and prevention strategies.

## Results

**Interactions between *C. jejuni* and *B. vulgatus*.** We assessed whether *C. jejuni* expresses fucosidase activity by incubating the strain with 4-nitrophenyl-α-L-fucopyranoside, a general α-L-fucosidase substrate that produces a yellow color when fucose is released. In contrast with *B. vulgatus*, *C. jejuni* is unable to cleave 4-nitrophenyl-α-L-fucopyranoside (Fig. 1a). Co-culture growth experiments were conducted to determine whether porcine mucin enhances *C. jejuni* growth in the presence of *B. vulgatus* and that was indeed what was observed. Furthermore, fucose metabolism contributed in part to this phenotype since a *C. jejuni fucP* mutant showed less growth enhancement under these conditions (Fig. 1b). In order to assess whether fucose was released from mucin by *B. vulgatus*, we isolated outer membrane vesicles (OMVs), since it is known that *Bacteroides* spp. release glycosidase-enriched OMVs[29], and co-incubated these with mucin to avoid bacterial consumption of the sugar. Free L-fucose was detected only when the *B. vulgatus* OMVs were present confirming that this carbon source can be released from mucin for *C. jejuni* use (Fig. 1c, d and Supplementary Fig. 1).

We further evaluated *C. jejuni*/*B. vulgatus* interactions by infecting germ-free mice with either a monoculture of *B. vulgatus* or a co-culture of *B. vulgatus* with *C. jejuni* and determined the impact of *C. jejuni* on *B. vulgatus* gene expression. Pathway enrichment analysis revealed that *B. vulgatus* upregulated numerous pathways involved in carbohydrate metabolism such as metabolic pathways bvu01100, carbon metabolism bvu01200 and microbial metabolism in diverse environments bvu01120 in the presence of *C. jejuni* (Supplementary Table 1). Furthermore, cecal and colon colony counts revealed that *B. vulgatus* levels remained constant whereas *C. jejuni* levels increased (although not to statistically relevant levels) in co-culture (Supplementary Fig. 2).

**Characterization of Cj0485 as a FabG homologue.** *Burkholderia multivorans* FabG (BmulJ_04919) is a monosaccharide dehydrogenase with activity on L-fucose, D-arabinose, and L-galactose[30]. Given that Cj0485 has 45% amino acid identity with FabG, we examined whether Cj0485 could also convert these sugars and/or others. Several aldose sugars in both D- and L-configurations were screened as substrates and Cj0485 displayed substantial activity on L-fucose and D-arabinose and with low levels of activity on L-galactose. Kinetic analysis of Cj0485 activity using an excess of L-fucose as a substrate revealed a preference for NADP$^+$ as a cofactor with $K_M$, $k_{cat}$, and $k_{cat}/K_M$ values of $9.9 \pm 1.9\,\mu M$, $16.5 \pm 0.7\,s^{-1}$ and $1.7 \pm 0.3\,\mu M^{-1}\,s^{-1}$, respectively compared with $K_M$, $k_{cat}$, and $k_{cat}/K_M$ values of $6050 \pm 1080\,\mu M$, $20.5 \pm 2.1\,s^{-1}$ and $0.0034 \pm 0.0008\,\mu M^{-1}\,s^{-1}$, respectively, for NAD$^+$. Using the preferred NADP$^+$ cofactor in excess, Cj0485 displayed ~4-fold selectivity for L-fucose with $K_M$, $k_{cat}$,

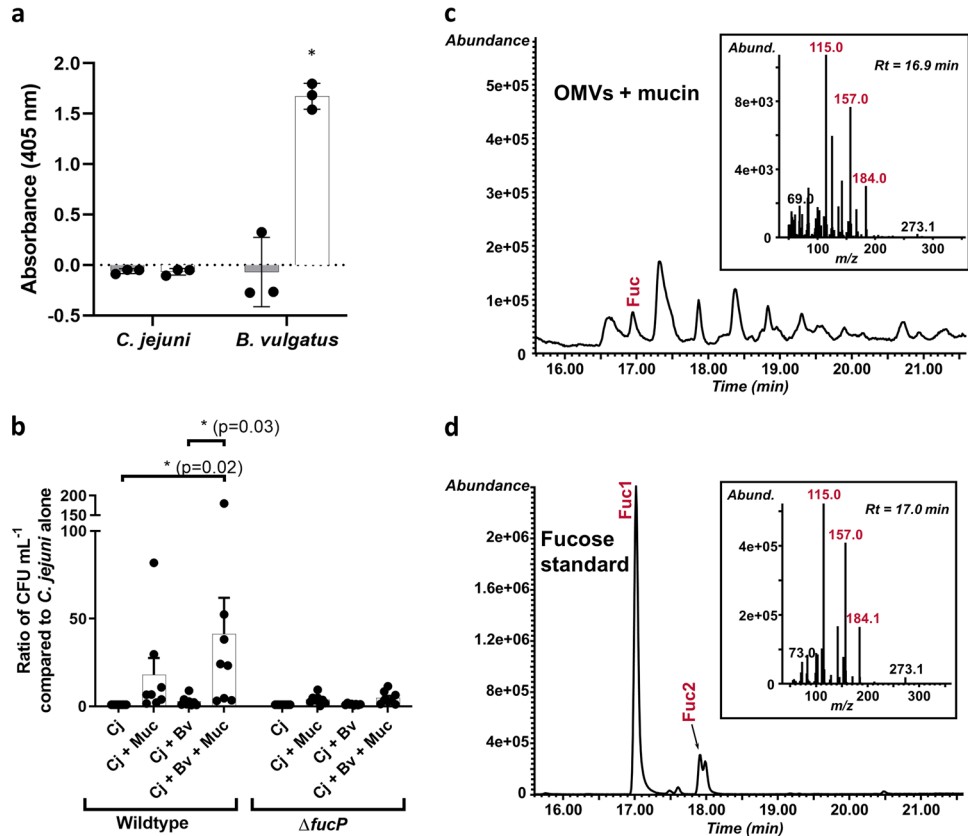

**Fig. 1 C. jejuni and B. vulgatus use of L-fucose and mucin degradation. a** Fucosidase activity detected by cleavage of 4-nitrophenyl-α-L-fucopyranoside measured by absorbance at 405 nm after 18 h incubation with the substrate under appropriate bacterial growth conditions (gray bars are at $t = 0$ h and white bars are at $t = 18$ h). Values represent means of three technical replicates and error bars show one standard deviation. Values from each replicate are overlaid as dots. The asterisk indicates a significant increase in absorbance ($p = 0.02$) compared to $t = 0$. **b** Ratios of C. jejuni (Cj) wildtype and *fucP* mutant in co-culture with B. vulgatus (Bv) and mucin (Muc) in minimal essential medium α. Values of bars represent means from eight biological replicates and error bars show one standard of the mean. Values from each replicate are overlaid as dots. Asterisks denote significance with *p*-values denoted in parentheses as determined by two-way ANOVA with Tukey post-test. **c** GC-MS chromatogram and spectrum (inset) of L-fucose released from mucin treated with B. vulgatus outer membrane vesicles (OMVs). **d** GC-MS chromatogram and spectrum (inset) of the L-fucose standard used to confirm the identity of L-fucose in the OMV-treated mucin sample. Fragment ions (Fuc1 and 2) are highlighted in red and exist in both α- and β- configurations and correspond to expected fragmentation patterns of fucose illustrated in Supplementary Fig. 1b.

and $k_{cat}/K_M$ values of 233.8 ± 26.2 μM, 26.3 ± 0.7 s$^{-1}$ and 0.11 ± 0.01 μM$^{-1}$ s$^{-1}$, respectively, compared with $K_M$, $k_{cat}$, and $k_{cat}/K_M$ values of 672.2 ± 82.8 μM, 18.9 ± 0.8 s$^{-1}$ and 0.028 ± 0.004 μM$^{-1}$ s$^{-1}$, respectively, for D-arabinose (Supplementary Fig. 3).

To provide insight into the substrate specificity of Cj0485, we solved X-ray crystal structures of this enzyme alone and in complex with NADP$^+$. The apo-enzyme crystallized in the space group C222$_1$ with two monomers in the asymmetric unit and the crystallographic lattice generates a tetramer with D2 symmetry. The holo-enzyme in complex with NADP$^+$ also crystallized in the space group P1. The holo-enzyme's asymmetric unit contained four monomers that were arranged as tetramers with identical D2 symmetry to the crystallographic tetramer observed in the apo-complex (Fig. 2a). PISA[31] analysis predicts this recurring tetramer to be a stable conformation, supporting the likelihood that the tetramer is the biologically relevant assembly of the enzyme.

The Cj0485 monomer adopts a Rossmann-like fold comprising a central seven-stranded twisted β-sheet sandwiched between two trios of parallel α-helices (Fig. 2b). A Cj0485 monomer from the apo-structure superimposes on an apo-FabG monomer with a root-mean-square-deviation (RMSD) of 0.9 Å (over 246 matched Cα atoms) revealing the similarity of the folds (Fig. 2c). Furthermore, the tetramer of Cj0485 superimposes on the tetramer of apo-FabG with an RMSD of 0.99 Å (over 1003

matched Cα atoms) indicating adoption of the same tetrameric organization.

Though we attempted extensive co-crystallization and soaking experiments, we were unable to obtain a ternary complex of Cj0485, NADP$^+$ and L-fucose. However, comparison of our NADP$^+$ complex to the ternary complex of FabG revealed insight into substrate specificity. Despite only 45% amino acid sequence identity, the NADP$^+$ and L-fucose binding sites are comprised of identical amino acid side chains, pointing to indistinguishable modes of substrate accommodation and consistent with the matching specificities of the two enzymes (Fig. 2d). The only notable difference, likely owing to the lack of bound fucose in Cj0485, was the more open conformation of the catalytic site in Cj0485 whereby an α-helical "lid" enclosing the L-fucose binding site was retracted by ~5.8 Å relative to its conformation in FabG. The monosaccharide binding site is organized specifically to provide hydrogen bonding with equatorial C1 (β-anomer), C2, and C3 hydroxyl groups of the typical $^1C_4$ chair conformation adopted by β-L-fucose (Fig. 2d, e). Likewise, the axial C4 hydroxyl also makes specific hydrogen bonding interactions while the position of W194, when the enzyme is in the closed lid conformation, would hinder accommodation of an axial C4 hydroxyl (Fig. 2d). These interactions are therefore tailored for β-L-fucose. Curiously, however, both enzymes have substantial

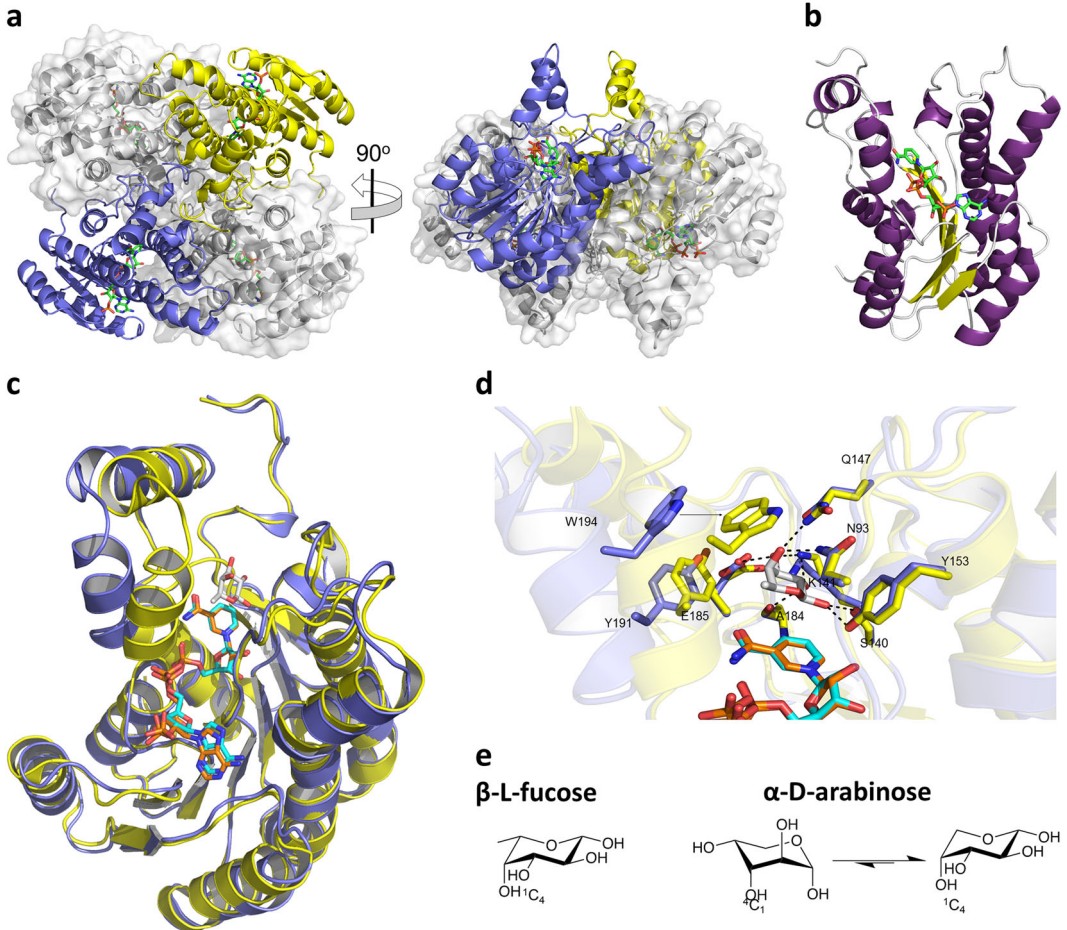

**Fig. 2 FucX crystal structure. a** The tetrameric organization of FucX observed in complex with NADP$^+$. Two monomers are shown in blue and yellow cartoon representation and the other two in grey with transparent solvent accessible surfaces. Bound NADP$^+$ is shown as green sticks. **b** Cartoon representation of the FucX monomer with β-strands shown in yellow, α-helices shown in purple, and loops shown in gray. Bound NADP$^+$ is shown as green sticks. **c** Structural alignment of *C. jejuni* FucX (in blue) in complex with NADP$^+$ (in cyan) with the homolog from *Burkholderia multivorans* (4GVX, in yellow) in complex with NADP+ (in orange) and L-fucose (in gray). **d** Close-up of the active site alignment of FucX (in blue) in complex with NADP+ (in cyan) with the homolog from *B. multivorans* (4GVX, in yellow) in complex with NADP+ (in orange) and L-fucose (in gray). Dashed lines indicate hydrogen bonds. E) Chair conformations of α-L-fucose and β-D-arabinose.

activity on D-arabinose. Notably, the α-anomer of D-arabinose favors the $^1C_4$ chair conformation[32], which has the same structural features as β-L-fucose, only lacking the C6 methyl group (Fig. 2e). Thus, we propose the activity of Cj0485 and FabG on D-arabinose is through selection of the $^1C_4$ chair conformation of α-D-arabinose from the conformational equilibrium of this monosaccharide. The apparent selectivity of these enzymes for β-L-fucose may result from van der Waals interactions between the enzyme active site and the C6 methyl group of this sugar, leading to a higher binding affinity compared with D-arabinose, which lacks the C6 methyl group. Data collection and structure statistics are available in Table 1. Due to our characterization of Cj0485 as a dehydrogenase and its previously known role in chemotaxis[25] we propose naming Cj0485 FucX and will use this name from here forward.

**Discovery of D-arabinose metabolism and chemotaxis.** Since FucX was active on D-arabinose, we tested whether *C. jejuni* could also metabolize and swim toward this sugar. It was found that *fuc* + strains showed enhanced growth in the presence of D-arabinose (Fig. 3a) and could swim to this attractant (Fig. 3b). Notably, expression of FucX without the rest of the *fuc* locus in the

*fuc*- 81–176 strain was sufficient to confer chemotaxis to both sugars (Fig. 3b). Furthermore, the growth advantage was lost in a fucose permease mutant and transport of radiolabelled fucose was reduced in the presence of D-arabinose, but not other similar aldoses (Fig. 3c), suggesting that both sugars are transported through the same permease. Additionally, the *fucX* mutant was also deficient in chemotaxis to D-arabinose (Fig. 3b), suggesting the dehydrogenase is also linked to recognition of this chemoattractant. *fuc* mutants that showed reduced growth on L-fucose[25] and $^3$H-fucose uptake (Supplementary Fig. 4) were also deficient in growth on D-arabinose (Fig. 3a) suggesting that the same enzymes involved in fucose catabolism are also involved in D-arabinose breakdown. We then performed NMR studies with $^{13}C_1$-L-fucose and $^{13}C_6$-L-fucose and detected L-fuconic acid (fuconate) (Supplementary Fig. 5). We also assessed if L-galactose could be metabolized since FucX showed very low levels of activity with this monosaccharide, but we could not detect any growth increase in its presence (Supplementary Fig. 6).

**Interplay between use of L-fucose and other carbon sources.** *C. jejuni* 11168 wildtype, together with the L-fucose permease mutant (*fucP*) and the *fuc* operon repressor mutant (*fucR*) were

**Table 1 Data collection and refinement statistics (molecular replacement).**

|  | FucX | FucX NADP complex |
|---|---|---|
| **Data collection** | | |
| Space group | C222$_1$ | P1 |
| Cell dimensions | | |
| $a, b, c$ (Å) | 101.2, 109.0, 105.7 | 68.5, 68.5, 70.7 |
| $\alpha, \beta, \gamma$ (°) | 90.0, 90.0, 90.0 | 89.3, 113.4, 115.7 |
| Resolution (Å) | 25.00–1.60 (1.63–1.60) | 30.0–2.15 (2.19–2.15) |
| $R_{merge}$ | 0.071 (0.328) | 0.116 (0.385) |
| $I / \sigma I$ | 22.9 (2.6) | 11.3 (2.4) |
| Completeness (%) | 99.7 (99.1) | 95.7 (95.7) |
| Redundancy | 4.7 (2.8) | 3.8 (2.2) |
| **Refinement** | | |
| Resolution (Å) | 1.60 | 2.15 |
| No. reflections | 69589 | 53537 |
| $R_{work} / R_{free}$ | 0.17/0.21 | 0.16/0.21 |
| No. atoms | | |
| Protein | 2074 (A), 2080 (B) | 2012 (A), 2040 (B), 2039 (C), 2022 (D) |
| Ligand/ion | 36 (EDO) | 192 (NAP), 18 (GOL), 4 (MG) |
| Water | 494 | 471 |
| *B*-factors | | |
| Protein | 18.6 (A), 17.3 (B) | 27.2 (A), 26.8 (B), 27.1 (C), 27.6 (D) |
| Ligand/ion | 24.3 (EDO) | 27.1 (NAP), 40.0 (GOL), 21.1 (MG) |
| Water | 28.7 | 30.9 |
| R.m.s. deviations | | |
| Bond lengths (Å) | 0.006 | 0.007 |
| Bond angles (°) | 0.745 | 0.956 |

Values in parentheses are for highest-resolution shell

screened on phenotypic arrays (Biolog) as an initial screen to determine if L-fucose metabolism influences the metabolism of other carbon sources. The *fucP* mutant was deficient in utilizing many alternative carbon sources as compared to the wild-type strain while the fucose regulator (*fucR*) mutant, which constitutively expresses the pathway[25], showed increased respiration on alternative carbon sources compared to the wildtype (Supplementary Fig. 7). Alternative carbon sources were also assessed for their ability to inhibit $^3$H-fucose uptake into the cell (Fig. 4). Excess L-serine, L-aspartic acid, and L-glutamic acid decreased $^3$H-fucose uptake, while no effect on L-fucose transport was observed when glucosamine, formic acid, proline, arginine, glutamine, asparagine, isoleucine or leucine were added to the growth medium (Supplementary Fig. 8). Quantitative RT-PCR (qRT-PCR) suggested that *fucP* expression was not substantially reduced in the presence of increasing amounts of L-serine, indicating that this regulation does not occur at the transcriptional level (Supplementary Fig. 9).

## Discussion

The ability to use L-fucose gives *C. jejuni* a competitive advantage when infecting a piglet model of diarrheal disease suggesting that fucose metabolism provides this foodborne pathogen with a fitness advantage in the host. Foraging carbohydrates from the human body has been linked to virulence in other organisms such as *Streptococcus pneumoniae*[33]. Interestingly, *S. pneumoniae* possesses two pathways for fucose degradation, but both rely upon cleavage of fucosylated structures in the respiratory tract rather than the import of free fucose[33] as does *C. jejuni*. In fact, *C. jejuni* 11168 has three identified glycosidases in the Carbohydrate-Active enZYmes database: MltD (SQF79585.1), SlT (SQF79837.1), and NCTC11168_01522 (SQF80505.1). MltD and SlT are important for maintaining cell wall integrity via peptidoglycan remodeling in the related organism *Helicobacter pylori*[34] while NCTC11168_01522 is a mannosyl-glycoprotein endo-beta-

N-acetylglucosamidase domain-containing protein, which has been described as an autolysin also important for peptidoglycan remodeling in *Staphylococcus aureus*[35]. It is likely these enzymes play similar roles in *C. jejuni* and are not fucosidases. Consistent with this, *C. jejuni* 11168 was unable to cleave the 4-nitrophenyl-α-L-fucopyranoside substrate (Fig. 1a). This substrate has been shown to be a general α-fucosidase substrate that can be cleaved by enzymes that also cleave α1–2, 1–3, 1–4, and 1–6 linkages although not all α-fucosidases will cleave it[36].

Because of the apparent inability of this bacterium to release L-fucose from glycans, we hypothesized that *C. jejuni* must forage for free fucose derived from the host diet or released from intestinal oligosaccharides (rather than cleaving the sugars itself) by other members of the gut microbiome as has been observed for enteric pathogens such as *S.* Typhimurium, *C. difficile*[24], and enterohemorrhagic *E. coli*[21]. We therefore designed co-incubation experiments with *B. vulgatus*, a member of a genus of gut microbes known to express multiple fucosidases[37] and preferentially packaging catalytic enzymes into secreted OMVs subsequently used to release nutrients for both itself and other gut bacteria to scavenge[29]. These experiments showed that mucin enhances *C. jejuni* growth when co-cultured with *B. vulgatus*, and this may in part be due to the release of free L-fucose from the mucin by *B. vulgatus*, since the *C. jejuni* fucose permease mutant did not show statistically increased growth under these conditions (Fig. 1b). To further validate these findings, we demonstrated that *B. vulgatus* OMVs released free L-fucose when incubated with the mucin used in the co-culture experiments (Fig. 1c, d and Supplementary Fig. 1). Previous research has indicated that *B. vulgatus* can provide nutrients such as xylan breakdown products to other microbiota members, but in turn relies on them for obtaining nutrients from other products including inulin, levan, and amylopectin[38], emphasizing the prevalence of nutrient sharing in the gut. Our in vitro results are also supported with gnotobiotic mouse colonization experiments revealing enhanced *C. jejuni* gut colonization in the presence of *B. vulgatus* (Supplementary Fig. 2). Although complex polysaccharides containing fucose are present in the mouse diets, carbohydrates are not the only carbon source available to the microbes as indicated by the variety of metabolic pathways upregulated by *B. vulgatus* when competing with *C. jejuni* for nutrients. Nonetheless, we observed a modest increase in *C. jejuni* colonization levels in the cecum and up to a fivefold increase in colonization in the colon when the organism was co-cultured with *B. vulgatus*. This suggests a tripartite interaction between *B. vulgatus, C. jejuni*, and the host whereby *B. vulgatus* may be providing nutrients for *C. jejuni* and may in turn also receive other nutrients from *C. jejuni*. For instance, a related *Bacteroides* species, *B. fragilis*, consumes human N-glycosylated molecules such as transferrin[39] and *C. jejuni* also produces N-glycans among other glycoconjugates[40], thus it is possible that *C. jejuni* glycans could provide a food source for *B. vulgatus*.

The enzymes encoded by the *C. jejuni* L-fucose locus are predicted to include a transcriptional regulator (Cj0480c/FucR), a synthase (Cj0481/DapA), a dehydratase (Cj0483), two major facilitator superfamily transporters (Cj0484 and Cj0486/FucP), two dehydrogenases (Cj0485/FucX and Cj0489), a hydrolase (Cj0487), and a mutarotase (Cj0488)[9]. Based upon the homology of the *C. jejuni fuc* operon-encoded enzymes to other characterized enzymes (such as those in the plant pathogen *Xanthomonas campestris*[41]), the growth and uptake phenotypes of the mutants tested in our previous[9,25] and current work, and the confirmed dehydrogenase activity of FucX, we developed a model for *C. jejuni* L-fucose and D-arabinose metabolism (Fig. 5). From NMR experiments using $^{13}$C-L-fucose, we detected large amounts of L-fuconic acid, which is consistent with our model and has

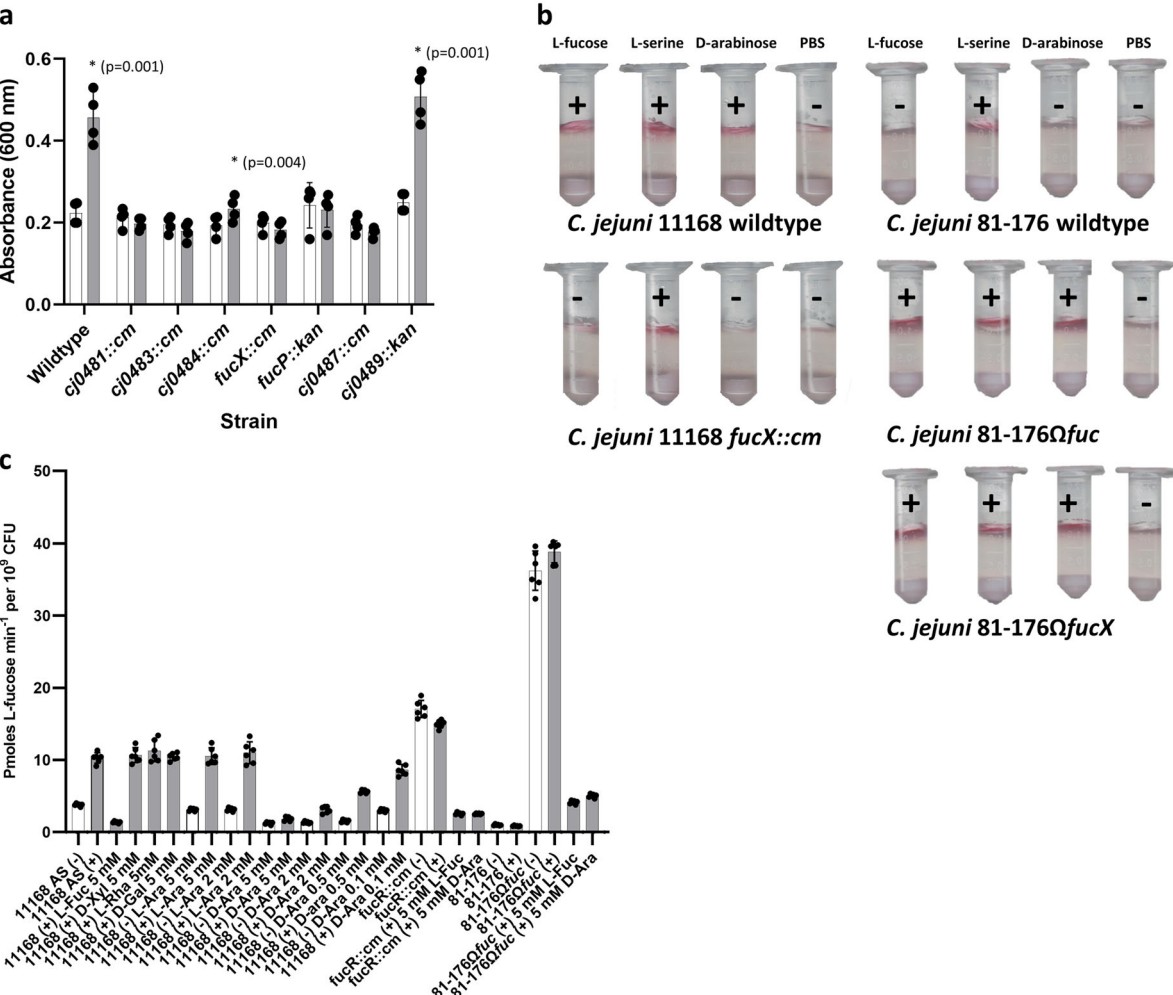

**Fig. 3 Phenotypic effects of ᴅ-arabinose. a** ᴅ-arabinose (25 mM) enhances wild-type C. *jejuni* 11168 growth (gray bars) relative to growth in unsupplemented minimal essential medium alpha (white bars) and this growth advantage is lost in the key metabolic mutants of the ʟ-fucose pathway. Values of bars represent means (n = 4 and error bars show one standard deviation). Values of individual replicates are overlaid as dots. Asterisks indicate a significant increase in growth in the presence of ᴅ-arabinose and *p*-values from a two-tailed paired Student's t-test are indicated in parentheses. **b** C. *jejuni* 11168 wild-type displays chemotaxis (indicated by pink color development and + sign) to ᴅ-arabinose and this is lost in a *fucX* mutant strain. Similarly, C. *jejuni* 81-176 wildtype does not show chemotaxis to ʟ-fucose and complementation with *fucX* confers this ability. ʟ-fucose serves as a FucX-dependent control, ʟ-serine serves as an independent positive control, and PBS serves as a negative control. **c** ᴅ-arabinose reduces transport of radiolabelled ʟ-fucose into fucose-utilizing cells (11168 wildtype and 81–176Ω*fuc*) more than other aldoses in both uninduced (white bars) and cells pre-grown with 20 mM ʟ-fucose (gray bars). Values represent means of two biological and two technical replicates, standard deviations are indicated by error bars, and values from individual replicates are overlaid as dots.

been observed by mass spectrometry studies[42]. Notably however, this latter work based their predicted pathway on the X. *campestris* pathway[42] as we had earlier predicted[9], but the current homology comparisons suggest a previously unrecognized model based upon the role of Cj0481 as a dihydrodipicolinate synthase. Interestingly, this pathway is identical to one recently described for the archaeon *Sulfolobus solfataricus*[43] but to our knowledge, this is the first time it has been identified in bacteria and it is distinctly different from that used by E.*coli*[44] or X. *campestris*[41].

The structural similarity of C. *jejuni* FucX to B. *multivorans* FabG (Fig. 2c) is consistent with the ability of both enzymes to efficiently reduce the same substrates, ʟ-fucose and ᴅ-arabinose, with weak activity on ʟ-galactose. FucX showed an ~4-fold preference for ʟ-fucose over ᴅ-arabinose (Supplementary Fig. 3) leading us to wonder whether ᴅ-arabinose and ʟ-galactose are true substrates in vivo. We did not observe enhanced growth with

ʟ-galactose (Supplementary Fig. 6), suggesting it is not a true substrate; however, C. *jejuni* growth increased in the presence of ᴅ-arabinose and mutants in the ʟ-fucose utilization locus were unable to grow on ᴅ-arabinose (Fig. 3b), suggesting it is a true substrate and catabolized by the same pathway as ʟ-fucose. Dual use of a fucose metabolic pathway for arabinose metabolism has been seen in other organisms including E. *coli*[44] and S. *sulfataricus*[45,46]; however, it remains to be determined if ᴅ-arabinose serves as a carbon source for these organisms in nature since ᴅ-arabinose is rare, although it has been identified in rosebuds[47] and recently in the alga, *Haematococcus pluvialis*[48].

Mutation of *cj0488* and *cj0489* did not impair C. *jejuni* growth. For the predicted Cj0488 mutarotase, we believe that since our assay conditions use excess ʟ-fucose in solution, there is enough fucose available in the desired beta configuration for the cell to catabolize the sugar without a mutarotase[49]. Normally mutarotases are required in vivo when there is a scarcity of water to

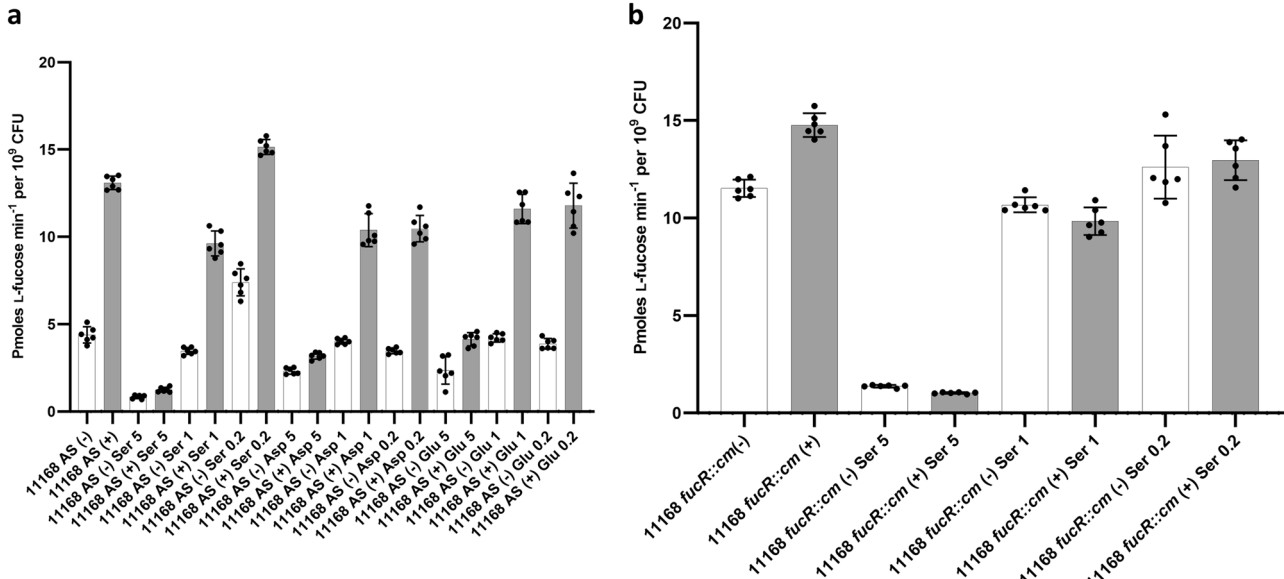

**Fig. 4 L-fucose uptake by *C. jejuni* 11168 grown in the presence of alternate carbon sources. a** $^3$H-L-fucose uptake rates in cells of the wildtype and **b** the *fucR::cm* mutant grown in medium supplemented with different concentrations (5 = 5:1 ratio supplement to fucose, 1 = 1:1 ratio supplement to fucose, 0.2 = 1:5 ratio supplement to fucose) of other carbon sources in the presence (gray) or absence (white) of L-fucose is shown. ser, L-serine; asp, aspartic acid; glu, glutamic acid. Each bar represents the mean value from two biological and two technical replicates, standard deviations are indicated by error bars.

catalyze the deprotonation and subsequent reprotonation that results in the configuration change[44]. For the putative Cj0489 dehydrogenase, it is possible that we would still observe enhanced growth since pyruvate is produced in the previous step serving as the primary energy source. Also, it is likely that the cell possesses a mechanism to cope with stress associated with excess aldehyde accumulation as observed in *E. coli*.

It is interesting to note that *C. jejuni* also displays chemotaxis to D-arabinose and that FucX is necessary for coordinating this response, as previously observed with L-fucose[25]. The mechanism for this response remains to be determined; however, we speculate that this, together with the *che* system, likely involves a direct or indirect interaction with one of the transducer-like proteins (Tlps) that serve as chemoreceptors in *C. jejuni*[50] since expression of FucX alone in the *fuc-* strain *C. jejuni* 81–176 is sufficient to confer chemotaxis to L-fucose and D-arabinose (Fig. 3b). Future work will focus on testing a collection of *C. jejuni* Tlp mutants for L-fucose and D-arabinose recognition to gain mechanistic insight into this coordinated response.

We compared the utilization of various carbon sources by wild-type *C. jejuni* cells versus *fucP* mutant cells (unable to uptake L-fucose) and *fucR* mutant cells (with enhanced L-fucose utilization due to constitutive expression of the operon) using the Biolog system. It was initially expected that the *fucP* mutant would have an enhanced ability to use other carbon sources; however, it was observed that the *fucP* mutant was deficient in the use of several of these sources, notably several tricarboxylic acid cycle intermediates such as succinate, malate, α-ketoglutarate, and pyruvate (Supplementary Fig. 7), suggesting that there may be a link between regulation of the fucose locus and these pathways. The inhibition of radiolabelled L-fucose uptake by other alternative carbon sources did not mirror the Biolog results completely, likely due to the different growth conditions necessary for the experiments. However, alternative carbon sources also influenced L-fucose uptake in this assay. For instance, L-serine, L-glutamic acid, and L-aspartic acid all reduced L-fucose uptake (Fig. 4). Catabolite-repression-based regulation of transcription

has previously been observed when examining *C. jejuni* use of amino acids such as serine and linked to intracellular levels of succinate[51]; however, our current work suggests no change in transcription of *fucP* (Supplementary Fig. 8), indicating that regulation is occurring via a different mechanism.

By better understanding the role of carbohydrate metabolism in *C. jejuni* and how other carbon sources and microbial species in the gut can influence this interaction, we may be able to better understand how nutrition influences *Campylobacter* carriage and disease severity. Future work will continue to build on these observations and may aid in the development of low-cost nutritional therapies.

## Methods

**Fucosidase activity assay**. Fucosidase activity was assessed based upon cleavage of 4-nitrophenyl-α-L-fucopyranoside (Gold Biotechnology)[52]. *C. jejuni* and *B. vulgatus* were incubated at $OD_{600} = 1$ in triplicate 10 μL volumes with 90 μL 1 M Tris-HCl, pH = 8.0 in 96-well plates with 10 mM 4-nitrophenyl-α-L-fucopyranoside dissolved in dimethylsulfoxide (DMSO) for 18 h under appropriate growth conditions. Absorbance at 405 nm was assessed at $t = 0$ and $t = 18$ h. Substrate-free and cell-free controls were included for all experimental groups. Significance was assessed by a two-tailed Student's *t*-test, assuming normal distribution.

**Co-culture experiments**. *C. jejuni* 11168 wildtype and *fucP::kan* and *B. vulgatus* were grown overnight on Mueller-Hinton (MH) or anaerobe plates under microaerobic or anaerobic conditions respectively. A $10^6$ colony-forming unit inoculum of *C. jejuni* was added to 2 mL of minimal essential medium α with glutamine and phenol red (Gibco), supplemented with 20 μM $FeSO_4$ and 0.5% porcine gastric mucin, and allowed to grow for 9 h under microaerobic conditions. Next, $10^7$ colony-forming units of *B. vulgatus* was added to the *C. jejuni* culture, and growth was continued for 24 additional hours under microaerobic conditions. A 30 μL aliquot was removed from the culture, serially diluted, and plated on Campy Line Agar[53] selective plates and incubated at 42 °C under microaerobic conditions. After 48 h, colonies were enumerated and statistical significance was determined by two-way ANOVA with Tukey post-test using GraphPad Prism 7.

**Detection of microbially-released L-fucose from mucin**. *B. vulgatus* were grown for 24 h under anaerobic conditions at 37 °C in 200 mL pre-reduced brain heart infusion broth supplemented with 5 μg mL$^{-1}$ hemin and 1 μg mL$^{-1}$ menadione. The culture was pelleted at 10,000 rpm for 15 min and the supernatant was filtered through a 0.45 μm filter and then a 0.22 μm PVDF membrane. The filtrate

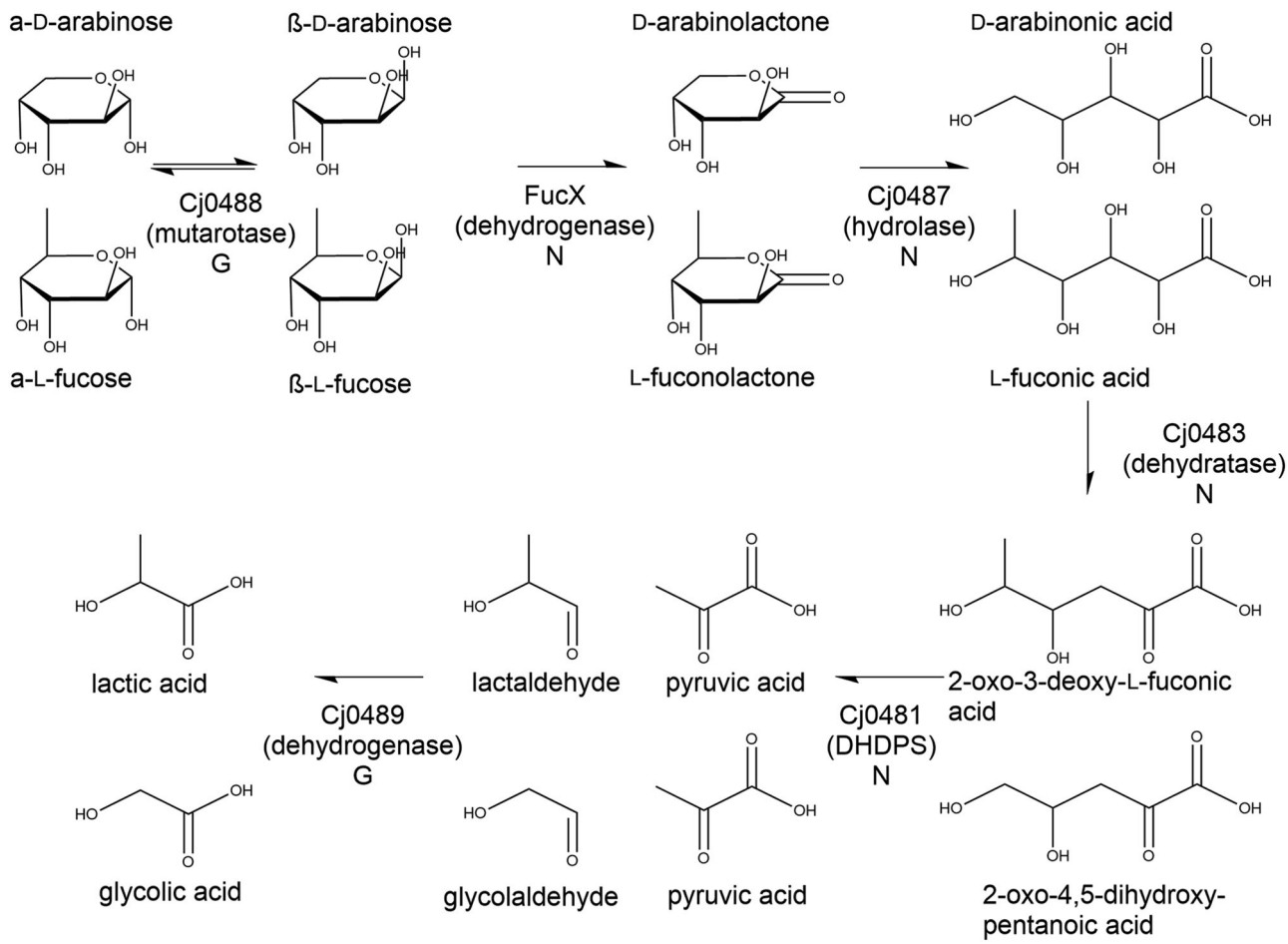

**Fig. 5 Predicted ʟ-fucose/ᴅ-arabinose metabolic pathway.** Model was developed based on homology to other characterized enzymes. The activity of FucX has been experimentally verified as has the production of ʟ-fuconic acid. Inner molecules trace the products formed by ʟ-fucose catabolism and outer molecules are from ᴅ-arabinose catabolism. The letters underneath each enzyme name denote the mutant growth phenotype (G = mutant has a growth enhancement on ʟ-fucose and ᴅ-arabinose, N = no growth enhancement as determined either in the present or previous[25] study).

was ultracentrifuged at $100{,}000 \times g$ for 2 h once, washed in 1X phosphate-buffered saline (PBS), and ultracentrifuged for an additional 2 h. The pellet containing the OMVs was then resuspended in 10 mL mqH$_2$O and quantified using a Pierce™ BCA Protein Assay Kit (Thermo Scientific) following the manufacturer's instructions. Twenty micrograms of OMVs were combined with 50 mM Tris-HCl, pH = 8.0 and 0.5% UV-sterilized porcine gastric mucin. Additional controls containing only OMVs or only mucin in buffer were also included. Samples were incubated at 37 °C with shaking at 90 rpm microaerobically for 18 h, then flash frozen in a dry ice/ethanol bath and lyophilized to dryness. In parallel, fucosidase activity of the OMVs was also verified by incubating with 4-nitrophenyl-α-ʟ-fucopyranoside as described for the whole cells. To detect free fucose released from mucin, the sample (OMVs + mucin) and controls (OMVs only, mucin only) were O-acetylated with acetic anhydride and pyridine at 45 °C for 25 min. Subsequently the sample and controls were analyzed by GC-MS on an Agilent 7890 A GC interfaced to a 5975 C MSD (mass selective detector, electron impact ionization mode). One microliter of sample was injected at a 100:1 split ratio. Separation was performed on a 30 m Supelco SP2380 column (30 m × 250 μm × 0.2 μm) using helium as carrier gas. The separation gradient started at 60 °C and this temperature was held for 1 min. The temperature was then increased to 170 °C at 27.5 °C min$^{-1}$, followed immediately by another increase to 235 °C, at 4 °C min$^{-1}$, and held for 2 min. A final ramp of 3 °C min$^{-1}$ raised the temperature to 240 °C and this temperature was held for 12 min. A post-run of 3 min at 250 °C was included after the runs for column cleaning.

**Gnotobiotic mouse mono and co-colonization.** Twelve female Swiss Webster germ-free mice were housed in pairs, in gnotobiotic isolators, where they were fed autoclaved standard chow diets (Teklad). The mice were divided into groups of six and inoculated with a single gavage of approximately 10$^7$ colony-forming units of B. vulgatus, or an equal mixture of both C. jejuni and B. vulgatus. Fourteen days post-inoculation, the mice were euthanized, weighed, and their cecum and colon were immediately collected. The protocol for the use of germ-free mice was

approved by the Institutional Animal Care and Use Committee of the University of Ottawa (Animal Care protocol BMI-2347).

To measure bacterial colonization, the samples were weighed, cut longitudinally, homogenized, serially diluted in fastidious anaerobe broth (Oxoid) and plated on MH (for C. jejuni) and anaerobe basal agar plates (Oxoid) (for B. vulgatus). The plates were incubated for 48 h under microaerophilic (for C. jejuni) or anaerobic (for B. vulgatus) conditions, and the resulting colonies were enumerated and expressed per gram of intestinal content. Statistical significance was tested using both t-tests and Mann–Whitney tests with GraphPad Prism 7.

To preserve RNA, whole mouse ceca (minus small samples removed for bacterial enumeration), were suspended in 5 mL of RNAlater (Ambion) and vortexed vigorously. The suspension was centrifuged briefly (15 s) at $500 \times g$ to remove debris. The remaining suspended bacteria and small particulate material was pelleted (15 min, $10{,}000 \times g$) and stored at −20 °C with fresh RNAlater.

**RNA extraction for RNAseq experiments.** Cultures were pelleted by centrifugation at $6000 \times g$ at 4 °C for 15 min. The pellets were resuspended and centrifuged again in 50% RNAlater/PBS buffer. Total RNA was extracted using a hot-phenol extraction method[54]. Genomic DNA was removed by two DNaseI (Epicentre) treatments, and the RNA was cleaned using RNeasy columns (Qiagen). A PCR was performed to ensure there was no genomic DNA present. Final RNA quality and quantity was ascertained using the BioRad's Experion StandardSense RNA system following the manufacturer's protocols. Ribosomal RNA was depleted from each sample using Terminator™ 5′-Phosphate-Dependent Exonuclease (Epicenter), using manufacturer protocols.

**Library construction for RNAseq.** Illumina-compatible cDNA libraries were constructed using a NEBnext® mRNA sample prep Master mix set 1 from 250 ng of rRNA-depleted total RNA using manufacturer protocols. Briefly, mRNA for each sample was fragmented, reverse transcribed to cDNA, dA-tailed and custom

designed adapters, each containing a specific barcode, were ligated to each cDNA fragment. The cDNA was size selected to 150–250 bp fragments using agarose gel extraction, and PCR enriched using adapter-specific primers. The samples were RNA extracted from the cecal contents of each gnotobiotic mouse for a total of 12 in vivo samples. The samples were sequenced on an Illumina HiSeq 2500.

**RNAseq data analysis**. Raw sequencing reads were demultiplexed using Novobarcode (Novocraft) and have been deposited to the NCBI SRA archive under accession number PRJNA514712. Each sample was aligned to a custom database containing both the *C. jejuni* NCTC 11168 (NC_002163.1) and *B. vulgatus* ATCC 8482 genomes (NC_009614.1) with Bowtie2[55] using local alignment and very-sensitive settings. Reads aligning to transcriptional regions were counted using HTSeq[56]. Gene differential expression and KEGG pathway enrichment analyses were performed[57–59]. *B. vulgatus* genes without a corresponding KEGG entry and pathways with no detectable expression were excluded from the pathway enrichment analysis. Results were considered significant with an FDR[60] corrected $p \leq 0.05$ (Please see Supplementary Table 1 for a list of exact *p*-values).

**Protein expression constructs**. UniProt IDs for all proteins mentioned in this paper are in Supplementary Table 2. All primers were synthesized by Eton Bioscience (Research Triangle Park, North Carolina) and are in Supplementary Table 3. Chromosomal DNA from *C. jejuni* 11168 served as template to amplify *fucX* with OneTaq DNA polymerase (New England Biolabs [NEB]) using primers Cj0485NdeIF and Cj0485XhoIR or Cj0485XhoIRc by PCR as recommended by manufacturer's instructions. Primers were designed to have XhoI and NdeI restriction sites for insertion into the pET28a-TEV and pET30a cloning vectors. Vectors and inserts were digested with XhoI (Promega) and NdeI (Thermo Fast Digest) following the ThermoScientific protocol for 75 min and purified using the Monarch PCR and DNA clean up kit (NEB) using the manufacturer's protocol. The fragment amplified with the Cj0485NdeIF primer and Cj0485XhoIR primer with the stop codon intact was ligated overnight at room temperature using T4 DNA ligase as per manufacturer's protocol into Xho/NdeI digested pET28a-TEV to create an N-terminally-His-tagged construct. The fragment with the alternative reverse primer (Cj0485XhoIRc) that disrupted the stop codon was ligated under similar conditions into pET30a to create a C-terminally-His-tagged construct. Following ligation, constructs were transformed into *E. coli* DH5α and confirmed by colony PCR using Taq polymerase (Invitrogen).

**FucX protein purification**. The pET expression constructs described in the previous section were isolated from DH5α using the GeneJET Plasmid Miniprep kit (Thermo Scientific) per manufacturer's instructions, modified to include 5 min drying prior to eluting with 30 μL 1/30 elution buffer pre-warmed to 55 °C and incubating for 15 min while eluting prior to spinning down. They were transformed into the BL21 expression strain of *E. coli*. A 5 mL overnight culture in 2x yeast extract and tryptone (YT) medium with kanamycin for plasmid maintenance was added to 250 mL fresh 2x YT and grown to an $OD_{600}$ of ~0.5 at 37 °C and 200 rpm shaking, then induced with 100 μL 1 M IPTG and the temperature reduced to 30 °C. After 2 h, cells were harvested by centrifugation at $4255 \times g$ for 30 min and frozen at −20 °C. The following day, pellets were thawed and passed twice through an EmulsiFlex™-C5 high pressure homogenizer in 10 mL lysis buffer (50 mM HEPES, 150 mM NaCl, 10% glycerol, 5 mM imidazole, pH = 7.5) per g pellet. For crystallography, the above protocol was scaled up to 4 L of 2x YT media and the harvested cells were disrupted by chemical lysis[61] instead of mechanical lysis. Debris was removed by centrifugation at $4255 \times g$ for 30 min, followed by an ultracentrifugation at $100,000 \times g$ for 1 h. Protein was isolated from the soluble fraction by nickel affinity chromatography using Ni-NTA columns (~ 0.75 cm × 0.75 cm resin bed), washing first with 10 column volumes mqH$_2$O and equilibrating with 20 column volumes lysis buffer then applying cell lysate and washing with 10 column volumes wash buffer (50 mM HEPES, 150 mM NaCl, 10% glycerol, 20 mM imidazole, pH = 7.5). Tagged protein was eluted in three fractions of three column volumes elution buffer (50 mM HEPES, 150 mM NaCl, 10% glycerol, 250 mM imidazole, pH = 7.5). For protein used in enzymology assays, the glycerol was omitted. Protein was desalted using PD-10 columns (GE Healthcare) per manufacturer's instructions and quantified by Bradford assay. Purified protein was stored in 100 μL aliquots at −20 °C.

**Crystallography**. Crystals were obtained at 18 °C using sitting-drop vapor diffusion for screening and hanging drop vapor diffusion for optimization. Crystals of purified C-terminally six-histidine tagged FucX (used at 50 mg mL⁻¹) were grown in 3.4 M 1,6-hexanediol, 0.1 M Tris pH 8.5 and 0.2 M magnesium chloride hexahydrate (MgCl$_2$ (H$_2$O)$_6$). Prior to flash freezing in liquid nitrogen, the crystals were cryoprotected in the crystallization solution supplemented with 25% (v/v) ethylene glycol. Crystals of a NADP⁺ complex were obtained by pre-incubation of the protein (at 30 mg mL⁻¹) with an excess of NADP⁺ followed by crystallization in 23–26% polyethylene glycol 4000, 0.08 M Tris pH 8.5, 0.15 M MgCl$_2$ (H$_2$O)$_6$ and 20% glycerol. These crystals were directly frozen in the crystallization solution.

Diffraction data were collected at 100 K on an 'in-house' beam comprising a Pilatus 200 K 2D detector coupled to a MicroMax-007HF X-ray generator with a VariMaxTM-HF ArcSec Confocal Optical System and an Oxford Cryostream 800.

Diffraction data were integrated, scaled and merged using HKL2000[62] and converted to a *mtz* file with SCALEPACK2MTZ[63]. The apo-structure of Cj0485 was solved by molecular replacement using PHASER[63] with the homolog from *B. multivorans* (4GVX) used as a search model. With these initial phases, BUCCANEER was used to automatically build a model of Cj0485[64], which was finished by manual building with COOT[65] and refinement with phenix.refine[66]. This complete model was used as a search model to solve the structure of the NADP⁺ complex. For both structures, the addition of water molecules was performed in COOT with FINDWATERS and manually checked after refinement. In all data sets, refinement procedures were monitored by flagging 5% of all observations as "free"[67]. Ramachandran statistics for both structures showed 96.5% preferred residues and 0.4% disallowed residues (corresponding to one residue in each monomer, V139). Model validation was performed with MOLPROBITY[68]. All data processing and model refinement statistics are shown in Table 1. The models obtained were represented using PyMOL (PyMOL Molecular Graphics System). Coordinates and structure factors for FucX and FucX in complex with NADP⁺ have been deposited with the PDB codes 6DRR and 6DS1, respectively.

**Enzymatic assay of FucX dehydrogenase activity**. An initial screen of 32 aldose sugars in both D- and L- configuration (glyceraldehyde, erythrose, threose, arabinose, lyxose, ribose, xylose, 5-deoxy-D-ribose, allose, altrose, galactose, glucose, gulose, idose, mannose, talose, and fucose) was performed by combining 1 μM dehydrogenase, 1 mM aldose sugar, 1 mM NAD⁺, and 1 mM Mg²⁺ in 50 mM Tris-HCl buffer (pH 7.5) and measuring the absorbance at 340 nm at 0, 6, and 30 min after the addition of Cj0485. Substrates with activity in the initial screen were used for further kinetic analysis.

All steady state kinetics were performed at room temperature on a SpectraMax M5 plate reader in UV transparent 96-well microtiter plates using SoftMax Pro 6.2.1 software. Standard reaction mixtures contained 50 mM Tris-HCl buffer (pH 7.5) and 1 mM Mg²⁺. These were supplemented with enzyme (1, 5, 10, and 200 nM), NAD⁺ (0–8 mM), NADP⁺ (0–400 μM), L-fucose (0–2.6 mM) or D-arabinose (0–3.6 mM). NADH and NADPH production was monitored at 340 nm; rates were calculated using the experimentally determined extinction coefficients, ε$_{340nm}$, of 5589.3 cm⁻¹ M⁻¹ and 5507.1 cm⁻¹ M⁻¹ for NADPH and NADH, respectively. All experiments were performed in triplicate. Michaelis-Menten parameters were determined by nonlinear curve fitting using GraphPad Prism.

**Growth and chemotaxis assays**. *C. jejuni* wildtype and *fuc* locus mutants at initial $OD_{600} = 0.05$ were grown in liquid culture for 18 h in minimal essential media α with glutamine and phenol red (Gibco), supplemented with 20 μM FeSO$_4$ with or without as 25 mM of the carbon source of interest (L-fucose, D-arabinose, or L-galactose) at 150 rpm and 37 °C under microaerobic conditions. Differences in final $OD_{600}$ between wildtype and mutant strains were compared using a two-tailed Student's *t*-test, assuming normal distribution, with four biological replicates. A tube based chemotaxis assay[25] was used to assess movement of 250 μL of *C. jejuni* 11168 wild-type and *fucX* mutant as well as 81–176 wildtype, Ω*fuc*, and Ω*fucX* in 0.4% molten PBS agar at a density of 2.8 mL g⁻¹ of cells from the bottom of a 2 mL tube through 1 mL sterile 0.4% PBS agar to a Whatman paper disc soaked in 50 μL 1 M L-fucose, D-arabinose, L-serine, or PBS after 3 days microaerobic incubation at 37 °C. Cells were detected by adding 200 μL of 0.01% triphenyltetrazolium chloride as an indicator and scored for presence (+) or absence (−) of pink color development after 1 h microaerobic incubation at 37 °C.

**Construction of mutant and complement strains**. The *C. jejuni* 11168 *cj0489::kan* mutant was constructed by the restriction digest method of insertion of a kanamycin cassette into the cloned gene of interest in the PCRscript vector, followed by electroporation, recombination, and selection for the mutant in *C. jejuni* grown on kanamycin-containing medium[25]. The cassette was inserted at a natural StyI restriction cut site located within the *C. jejuni* genome at bases 455,818–455,103. The *C. jejuni* 81-176Ω*fucX* complement strain was constructed by electroporation of genomic DNA from a previously constructed[25] complement strain containing *fucX* into the parent strain.

**NMR analysis of cellular metabolites**. *C. jejuni* 11168 wild-type, *cj0488*, *cj0481*, and *cj0489* mutants were grown on MH plates for two days and subcultured on MH plates for one day further after which 50 mL minimal essential medium α with glutamine and phenol red (Gibco), with 20 μM FeSO$_4$ as used in growth experiments was inoculated to an $OD_{600} = 0.35$ and grown overnight in microaerobic conditions at 37 °C and 150 rpm. The following morning, 10 mL of the culture was added to 10 mL of fresh growth medium containing either 50 mM of a 9:11 ratio of ¹³C-1-L-fucose to ¹³C-6-L-fucose or unsupplemented and incubated for 8 h. Cultures were then pelleted by centrifugation at $4255 \times g$, washed twice with 4 mL cold PBS, and flash frozen. Cells were lysed and metabolites extracted using the method described by Soo et al[69], modified to filter using VIVASPIN 20 10 K MWCO columns (Sartorius) prewashed eight times with distilled water to remove glycerol and the filtrate lyophilized. Lyophilized extract was resuspended in 99.96% D$_2$O. Spectra were acquired on a Bruker Neo 800 MHz spectrometer equipped with a 5 mm HCN cryoprobe. Standard Bruker pulse sequences were used (HSQC: hsqcetgpsisp2, HSQC-TOCSY: hsqcetgpml, HMBC: hmbclpndqf). Processing was

done with Mestrelab MNOVA software. Chemical shifts were referenced to internal DSS. An authentic sample of L-fuconic acid to verify the proton and carbon shifts and identify the compound in the extract.

**Biolog screening**. Carbon source utilization was screened for *C. jejuni* 11168 wildtype, *fucR::cm*, and *fucP::cm* on Biolog PM1 and PM2A phenotypic microarray plates, each containing 95 unique carbon sources. Cells were added to the arrays at 52% transmittance and incubated microaerobically in an OmniLog PM at 37 °C for 72 h with reads every 15 min.

**$^3$H-L-fucose uptake assays**. Wildtype or mutant cells were grown in the presence or absence of L-fucose and then tested for their ability to take in $^3$H-L-fucose when supplemented with L-serine, aspartic acid, and D- and L-arabinose, glutamic acid, glucosamine, formic acid, proline, arginine, glutamine, asparagine, isoleucine or leucine. Overnight cultures of cells were adjusted to $OD_{600} = 1.0$ and used to inoculate 10 mL of MH (supplemented with 25 mM L-fucose and/or alternative carbon source as needed) with 2 mL of cells. The 12 mL cultures were grown under microaerophilic conditions with shaking (100 rpm) at 37 °C until mid-log phase (~4 h). At this point, the $^3$H-L-fucose was added and its uptake measured by scintillion oscillography for radioactivity of cells harvested from the culture. Significance was determined by a two-tailed Student's *t*-test *p*-value lower than 0.05, assuming normal distribution.

**qRT-PCR**. The expression of *fucP* was compared in cells grown in differing L-fucose and L-serine supplement conditions. These conditions included unsupplemented, excess serine (50 mM serine, 10 mM fucose), equimolar (10 mM of each), excess fucose (10 mM fucose, 2 mM serine), and serine only and fucose only (10 mM each). Cells were initially thawed on MH plates, restreaked after 36 h and grown for 24 h microaerobically at 37 °C. These cells were then harvested in MH broth and set to $OD_{600} = 1.0$. Two milliliters of this solution were then used to inoculate 10 mL of MH broth (with supplements when needed). Cultures were grown under microaerobic conditions at 37 °C with shaking at 100 rpm until mid-log phase (~4 h), after which cells were harvested and RNA extracted using the RNeasy Mini Kit (Qiagen) per the manufacturer's instructions for bacterial cells with lysozyme-mediated lysis. Genomic DNA was removed by treatment with TURBO DNA-free$^{TM}$ kit per manufacturer's instructions. Reverse transcription was carried out using 800 ng of RNA with iScript$^{TM}$ Reverse Transcription Supermix for RT-qPCR. Quantitative PCR was carried out using the Bio-Rad SsoAdvanced$^{TM}$ Universal SYBR® Green Supermix kit per manufacturer's instructions with 0.4 μM *cj0486* primers (cj0486-RT-F and cj0486-RT-R) or 0.4 μM 16S control primers (16s-RT-For and 16s-RT-Rev) described in Table 1 in a Bio-Rad C1000 Touch$^{TM}$ CFX96$^{TM}$ Real-Time System, and data were analyzed using Bio-Rad CFX Manager$^{™}$ Software 3.1. Reactions were carried out in two biological replicates and significance was determined using a two-tailed Student's *t*-test assuming normal distribution.

**Statistics and reproducibility**. All statistical tests performed are indicated in the relevant methods above. All experiments were reproduced at least once unless otherwise indicated.

**Reporting summary**. Further information on research design is available in the Nature Research Reporting Summary linked to this article.

## Data availability

All relevant data are available from the corresponding author. The source data underlying Figs. 1a, b, 3a–c and 4a, b and Supplementary Figs. 2, 3, 4, 6, 7, 8, and 9 are provided in Supplementary Data 1. The spectral data for Supplementary Figs. 1 and 5 are provided Supplementary Data 2. The RNAseq data has been deposited under accession number PRJNA514712. Coordinates and structure factors for FucX and FucX in complex with NADP+ have been deposited with the PDB codes 6DRR and 6DS1, respectively. Their PDB validation reports are provided as Supplementary Data 3 and 4, respectively.

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

## Acknowledgements

The authors would like to thank David Kelly, Victoria Korolik, Evgeny Vinogradov and Hanwen Huang for helpful discussions. CMS is an Alberta Innovates Strategic Chair in Bacterial Glycomics. This research was supported by Discovery (RGPIN 2014-04355) from the Natural Sciences and Engineering Research Council of Canada to ABB and the National Institutes of Health funded Research Resource for Biomedical Glycomics (P41GM10349010) to PA. Research in the AS laboratory is supported by CIHR grant MOP#84224.

## Author contributions

J.M.G., H.N., and C.M.S. designed the study with insight from A.B.B., A.S., P.A., and J.A.G. J.M.G. cloned, expressed, and purified proteins. B.P. carried out X-ray crystallography and FucX enzyme kinetics. J.M.G. and H.H. performed preliminary FucX enzyme activity screens. J.M.G. completed all growth and chemotaxis experiments with assistance from X.B. and A.E. J.M.G. purified cellular metabolites for NMR analysis done by J.G. X.B. created the *cj0489* mutant strain with guidance from J.M.G. S.P. completed the GC-MS experiments. J.M.G. performed Biolog experiments with guidance from E.L. H.N. designed and completed the uptake assays. X.B. and J.M.G. prepared the cultures and did the qRT-PCR experiments. J.M.G. performed the fucosidase activity assays. J.M.G. completed the co-culture experiments with guidance and data analysis from M.S. M.S. performed the mouse studies and J.B. analyzed the RNA-seq data. J.M.G. and C.M.S. wrote the paper with input from all authors.

## Competing interests

The authors declare no competing interests.
