## [Peer Review File · Communications Biology]

Reviewers' comments:

Reviewer #1 (Remarks to the Author):

The manuscript investigates the molecular basis of fucose metabolism by *Campylobacter jejuni* and its role in chemotaxis and colonization.

The key findings are based on the enzymatic and structural characterization of a dehydrogenase (FabG) which the authors show to be a critical enzyme in the metabolic pathway of *C. jejuni* fucose utilization, as previously identified in *Burkholderia multivorans*. Confirmation of the importance of the fucose operon in *C. jejuni* chemotaxis, cross-feeding and colonization was shown in vitro using a series of *C. jejuni* mutants and in vivo using gnotobiotic mice.

The experimental approach is sound, the manuscript clearly written, and the findings significantly advance the field of research.

Major comments

1. Regarding the in vitro co-culture experiments, it should be noted that a number of microbial fucosidases while not active on pNP-fucose, can be active against different fucosylated oligosaccharides or glycoconjugates. Have/could the authors tried to assess *C. jejuni* fucosidase activity using a range of fucosylated substrates, in support of their findings on pNP-fucose?
2. When growing *C. jejuni* and *B. vulgatus* on porcine mucin, could the authors monitor fucose in the supernatant by NMR to substantiate the hypothesis that *C. jejuni* benefits from fucose released from mucin by *B. vulgatus*?
3. Discussion. The authors should revise the sentence page 13 referring to *Streptococcus pneumoniae* as an example of pathogen linked to the ability of foraging on carbohydrates in the gut, as *Streptococcus pneumoniae* is a colonizer of the upper airways. *Salmonella* or *E. coli* would be a better example.
4. Given the homology of FabG to *Burkholderia multivorans*, could the authors discuss the relevance of fucose metabolism of this species in the Discussion section?

Minor comments

1. Abstract- Could the authors clarify the gnotobiotic mice. It is not clear from the sentence that these are co-colonized with both *C. jejuni* and *B. vulgatus*.
2. Introduction- The authors should introduce at the start what fuc+ strains refers to, I believe it is strain-positive for the fucose operon, but it may be interpreted differently (fucosidase positive, fucose growing positive etc).
3. There seems to be issues with reference numbering. For example, ref #4 does not seem to be the correct one when referring to *Campylobacter succinate* (lane 268 page 13) or *Streptococcus pneumoniae* (lane 272 page 13).

Reviewer #2 (Remarks to the Author):

In this manuscript the authors studied the metabolism of fucose in *Campylobacter jejuni*. It is a follow up on the work by Stahl et al PNAS 2011 "L-Fucose utilization provides *Campylobacter jejuni* with a competitive advantage". The authors characterized "Cj0485 through crystallography, enzymology and multiple biological assays to confirm its function as a fucose dehydrogenase, which [they] subsequently name FabG. The *C. jejuni* FabG is also capable of reducing D-arabinose and is the sole component encoded by the fuc operon necessary for chemotaxis to both sugars. Nuclear magnetic resonance (NMR) and mutagenesis studies provide further insight into the mechanism of L-fucose breakdown and allowed [them] to propose a pathway for L-fucose and D-arabinose catabolism. [They] then investigate the impact of L-fucose on carbon source preferences by *C. jejuni* and examine the crosstalk that may occur between commensal microbes and this pathogen."

The novelty of the results is a deeper and more detailed understanding of the fucose metabolism of *C. jejuni*.

The results would interest the community of scientists working with microbes related to gut health. *C. jejuni* is a pathogen of the gut and the fact that it uses fucose makes it a competitor for other bacteria using fucose in the gut.

The authors did a thorough and detailed analysis. The methods and statistical analyses are valid.

I would suggest maybe reshuffling the text so the broader impact (the fact that *C. jejuni* is a gut pathogen and that it competes for fucose with beneficial microbes in the gut) is earlier in each section such as the abstract and the introduction (for ex. the last sentence of the abstract only mentions that *C. jejuni* is a pathogen). Also, why fucose metabolism is important is fascinating but only explained in the discussion. I would suggest mentioning this earlier than in the discussion.

Minor comments:

Page 9 Lines 192-193: The way the sentence is written should be changed to clarify, for example "Metabolic pathways (bvu01100)", should be written "metabolic pathways bvu01100", otherwise it sounds like Metabolic pathway is in itself a pathway. (?) Same for the rest of the sentence: "Carbon metabolism (bvu01200)", "Microbial metabolism in diverse environments (bvu01120)"... from that sentence: "Pathway enrichment analysis revealed that *B. vulgatus* upregulated numerous pathways involved in carbohydrate metabolism such as Metabolic pathways (bvu01100), Carbon metabolism (bvu01200) and Microbial metabolism in diverse environments (bvu01120) in the presence of *C. jejuni*"

p17 line 364: Replace "VariMaxTM-HF Arc)Sec" by "VariMaxTM-HF ArcSec"

Elsa Petit

Comments for Reviewers

Referee expertise:

Referee #1: Glycobiology of host-microbe interactions

Referee #2: Bacterial utilization of fucose

Note to reviewers: It was brought to our attention that FabG is traditionally used for lipid metabolism, therefore we have changed the name of our enzyme to FucX throughout the manuscript to avoid confusion. The X designation is to avoid overlap with other Fuc enzymes and also to indicate the association with chemotaXis.

Reviewers' comments:

Reviewer #1 (Remarks to the Author):

Major comments

1. Regarding the in vitro co-culture experiments, it should be noted that a number of microbial fucosidases while not active on pNP-fucose, can be active against different fucosylated oligosaccharides or glycoconjugates. Have/could the authors tried to assess *C. jejuni* fucosidase activity using a range of fucosylated substrates, in support of their findings on pNP-fucose?

pNP-fucose has been shown to be a general fucosidase substrate that can be cleaved by enzymes that also cleave α 1-2, 1-3, 1-4, and 1-6 linkages although the reviewer is correct that not all fucosidases will cleave it (Lezyk et al. 2016). Also, *C. jejuni* 11168 has three identified glycosidases in the Carbohydrate-Active enZymes (CAZy) database – MltD (SQF79585.1), SIT (SQF79837.1), and NCTC11168_01522 (SQF80505.1). MltD and SIT are important for maintaining cell wall integrity via peptidoglycan remodeling in the related organism, *Helicobacter pylori* (Chaput et al. 2007), so it is likely this is also their function in *C. jejuni*. NCTC11168_01522 is a mannosyl-glycoprotein endo-beta-N-acetylglucosamidase domain-containing protein, which has been described as an autolysin in *Staphylococcus aureus* also important for peptidoglycan remodeling (Oshida et al. 1995). We have adjusted the discussion of these results to include this information for clarity.

2. When growing *C. jejuni* and *B. vulgatus* on porcine mucin, could the authors monitor fucose in the supernatant by NMR to substantiate the hypothesis that *C. jejuni* benefits from fucose released from mucin by *B. vulgatus*?

We thank the reviewer for this excellent suggestion. We have done the experiment using mass spectrometry as a more sensitive detection method and because our mucin is not ¹³C-labeled. We have demonstrated that fucose is released from porcine mucin by outer membrane vesicles (OMVs) harvested from *B. vulgatus* since it is known that fucosidases from *Bacteroides* spp. are present in OMVs (Elhenawy et al. 2014) and to minimize bacterial consumption of fucose prior to detection of the metabolite. We believe this further substantiates the hypothesis that released fucose can indeed serve as a nutrient for *C. jejuni*. We have added these results into Figure 1cd and Supplementary Figure S1. The experiment is described in the Methods, Results and Discussion sections. We have also added two additional authors, Sara Porfirio and Parastoo Azadi who were involved in doing the mass spectrometry for this experiment (and their funding source was added to the acknowledgements).

3. Discussion. The authors should revise the sentence page 13 referring to *Streptococcus pneumoniae* as an example of pathogen linked to the ability of foraging on carbohydrates in the gut, as *Streptococcus pneumoniae* is a colonizer of the upper airways. *Salmonella* or *E. coli* would be a better example.

We agree and thank the reviewer for this observation and have expanded the section to discuss *S. pneumoniae* as a glycan forager in other parts of the body and include both suggested examples as pathogens present in the gut.

4. Given the homology of FabG to *Burkholderia multivorans*, could the authors discuss the relevance of fucose metabolism of this species in the Discussion section?

To our knowledge, there has only been one publication on fucose metabolism by *B. multivorans*, Hobbs et al, 2013, which we cite in our manuscript. This publication focuses solely on the biochemical characterization of the pathway and thus nothing is known regarding the biological relevance of this pathway – or whether it is even functional in *B. multivorans*. One could speculate the pathway likely has similar roles in nutrient acquisition during host infection as we discussed for the other organisms, but we did not think it was appropriate for us to speculate on the relevance for *Burkholderia*.

Minor comments

1. Abstract- Could the authors clarify the gnotobiotic mice. It is not clear from the sentence that these are co-colonized with both *C. jejuni* and *B. vulgatus*.

Yes, we have made the suggested clarification.

2. Introduction- The authors should introduce at the start what fuc+ strains refers to, I believe it is strain-positive for the fucose operon, but it may be interpreted differently (fucosidase positive, fucose growing positive etc).

We have corrected this section for clarity.

3. There seems to be issues with reference numbering. For example, ref #4 does not seem to be the correct one when referring to *Campylobacter succinate* (lane 268 page 13) or *Streptococcus pneumoniae* (lane 272 page 13).

This was a citation program error and we thank the reviewer for pointing it out. It has been corrected throughout the manuscript.

Reviewer #2 (Remarks to the Author):

I would suggest maybe reshuffling the text so the broader impact (the fact that *C. jejuni* is a gut pathogen and that it competes for fucose with beneficial microbes in the gut) is earlier in each section such as the abstract and the introduction (for ex. the last sentence of the abstract only mentions that *C.jejuni* is a pathogen). Also, why fucose metabolism is important is fascinating but only explained in the discussion. I would suggest mentioning this earlier than in the discussion.

We thank the reviewer for this good suggestion. We have rearranged the order in the Introduction, Results, Figures, Discussion and Methods of the manuscript as suggested to emphasize these points. We struggled to change the order in the abstract without going beyond the 150 word limit so we moved the pathogen statement to the first sentence to emphasize the importance of our organism and hope the rest is satisfactory to the reviewer.

Minor comments:

Page 9 Lines 192-193: The way the sentence is written should be changed to clarify, for example "Metabolic pathways (bvu01100)", should be written "metabolic pathways bvu01100", otherwise it sounds like Metabolic pathway is in itself a pathway. (?) Same for the rest of the sentence: "Carbon metabolism (bvu01200)", "Microbial metabolism in diverse environments (bvu01120)"... from that sentence: "Pathway enrichment analysis revealed that *B. vulgatus* upregulated numerous pathways involved in carbohydrate

metabolism such as Metabolic pathways (bv01100), Carbon metabolism (bv01200) and Microbial metabolism in diverse environments (bv01120) in the presence of *C. jejuni*"

We have made those suggested changes.

p17 line 364: Replace "VariMax™-HF Arc)Sec" by "VariMax™-HF ArcSec"

We have made this change.

REVIEWERS' COMMENTS:

Reviewer #1 (Remarks to the Author):

The authors satisfactorily answered the comments. The addition of the OMV experiment is welcomed and provides added value to the study.